# Comprehensive Comparison of 22C3 and SP263 PD-L1 Expression in Non-Small-Cell Lung Cancer Using Routine Clinical and Conditioned Archives

**DOI:** 10.3390/cancers14133138

**Published:** 2022-06-27

**Authors:** Sue Youn Kim, Tae-Eun Kim, Chan Kwon Park, Hyoung-Kyu Yoon, Young Jo Sa, Hyo Rim Kim, In Sook Woo, Tae-Jung Kim

**Affiliations:** 1Department of Hospital Pathology, Seoul St. Mary’s Hospital, College of Medicine, The Catholic University of Korea, Seoul 06591, Korea; sueyun818@catholic.ac.kr; 2Department of Hospital Pathology, Yeouido St. Mary’s Hospital, College of Medicine, The Catholic University of Korea, 10, 63-ro, Yeongdeungpo-gu, Seoul 07345, Korea; lovelyposh7@gmail.com; 3Division of Pulmonology, Department of Internal Medicine, Yeouido St. Mary’s Hospital, College of Medicine, The Catholic University of Korea, 10, 63-ro, Yeongdeungpo-gu, Seoul 07345, Korea; ckpaul@catholic.ac.kr (C.K.P.); cmcyhg@catholic.ac.kr (H.-K.Y.); 4Department of Thoracic Surgery, Yeouido St. Mary’s Hospital, College of Medicine, The Catholic University of Korea, 10, 63-ro, Yeongdeungpo-gu, Seoul 07345, Korea; sagoon@catholic.ac.kr; 5Department of Radiology, Yeouido St. Mary’s Hospital, College of Medicine, The Catholic University of Korea, 10, 63-ro, Yeongdeungpo-gu, Seoul 07345, Korea; horrim@catholic.ac.kr; 6Department of Medical Oncology, Yeouido St. Mary’s Hospital, College of Medicine, The Catholic University of Korea, 10, 63-ro, Yeongdeungpo-gu, Seoul 07345, Korea; insookwoo@catholic.ac.kr

**Keywords:** lung cancer, non-small-cell lung cancer, programmed death ligand 1, 22C3, SP263, tumor proportion score

## Abstract

**Simple Summary:**

The PD-L1 22C3 and SP263 assays are routinely used to select patients for immunotherapy in NSCLC, and their IHC expression results demonstrate a strong correlation. However, their interchangeability should be carefully examined when using long-term preserved FFPE blocks more than 3 years or paraffin sections held for more than one week at room temperature conditions, where the PD-L1 sensitivity of SP263 is superior to that of 22C3.

**Abstract:**

PD-L1 harmonization studies revealed a strong correlation between the 22C3 and SP263 assays in non-small-cell lung cancer (NSCLC). However, the assays’ characteristics have yet to be validated in a variety of clinical and analytical settings. The results of 431 NSCLC samples tested concurrently in routine clinical practice with the PD-L1 22C3 and SP263 assays were reviewed, and both assays were performed on 314 archives of surgically resected NSCLCs to assess PD-L1 expression in relation to variables such as FFPE block age and FFPE section storage condition. In routine clinical samples, 22C3 showed the highest concordance rate with 94.5% of SP263 tumor proportion score (TPS) ≥50% and 92.3% of SP263 TPS ≥1%, while SP263 showed a concordance rate with 79.6% of 22C3 TPS ≥50% and 89.9% of 22C3 TPS ≥1%. In the archival analysis, the high TPS of 22C3 and SP263 (versus TPS 1%) were significantly associated with a more recent block (<3 years versus ≥3 years) (p = 0.007 and p = 0.009, respectively). Only the TPS of 22C3 was reduced when FFPE sections were stored at room temperature compared to SP263. However, when stored at 4 °C, the storage duration had no effect on expression in either assay. For 22C3 TPS 1–49 percent and ≥50 percent (OR = 1.73, p = 0.006 and OR = 1.98, p = 0.002, respectively). There was a considerably larger chance of preserved 22C3 expression in recent room-temperature paraffin section storage, although SP263 demonstrated preserved expression in prolonged room-temperature section storage. Despite the good association between PD-L1 22C3 and SP263 in routine clinical samples, FFPE blocks older than 3 years and sections held at room temperature for more than 1 week may result in an underestimation of PD-L1 status, particularly for the 22C3 test. However, the SP263 assay was more sensitive under these conditions.

## 1. Introduction

The introduction of programmed death receptor 1 (PD-1)/programmed death ligand 1 (PD-L1) inhibitors changed cancer immunotherapy, demonstrating an overall survival improvement over chemotherapy with fewer side effects [1,2,3,4,5,6]. PD-1/PD-L1 inhibitors such as pembrolizumab, nivolumab, atezolizumab, and durvalumab are currently approved by the United States Food and Drug Administration. For patient selection, each agent had its own customized PD-L1 assay. There are four anti-PD-L1 clones: 22C3, 28-8, SP263, and SP142 assays. Attempts to overcome the “one-drug, one assay” paradigm are being made worldwide for more efficient treatment decision-making. As these are four different assays targeting the same pathway, attempts to harmonize these diagnostics are in progress [7,8,9]. A blueprint PD-L1 IHC assay interchange feasibility project and subsequent studies have suggested high comparability in 22C3 and SP263 [7,10,11,12].

However, some extent of discrepancy between different assays has been proposed [9]. IHC interpretation of PD-L1 expression has solely relied on formalin-fixed, paraffin-embedded tissue (FFPE). In the context of histopathological diagnoses and the search for immunotherapy candidates, FFPE blocks are valuable resources. These archives are a valuable resource for PD-L1 evaluation and the development of new immuno-oncology diagnostics. The vast majority of FFPE tissue is stored at room temperature, and it has been widely documented that tissue stored at room temperature for decades can still be used for histological examination. Furthermore, despite increasing levels of degradation, protein, and nucleic acids can be retrieved from archives, allowing for molecular studies such as tumor mutational load and PD-L1 expression status [13,14]. However, no systematic evaluation of the effect of long-term storage conditions on the quality of PD-L1 expression in FFPE tissues has been conducted. Moreover, such studies do not exist at various FFPE section storage conditions.

Quality assurance recommendations for PD-L1 IHC preanalytical, analytical, and post-analytical parameters such as a short ischemia period, avoidance of decalcification, and the necessity for internal and external controls were proposed by the International Association for the Study of Lung Cancer (IASLC) [11]. While PD-L1 tests can be employed in a range of experimental and clinical contexts employing FFPE tissues with pathologic or clinical value, a thorough assessment of the effect of storage conditions on the quality and integrity of biomolecules in FFPE tissues is currently absent. Because immunotherapy is only approved for persons with advanced cancer, clinicians often take a little sample from a metastatic site to make the diagnosis. As a result, strict compliance with the aforementioned recommendations necessitates additional sampling to determine the optimal condition, resulting in multiple sampling, which causes delays, costs, and potential complications and is frequently impossible due to factors such as tumor location and patient functional status [15]. Storage variables such as FFPE block age, long-term paraffin section storage, and poor storage conditions are all examples of improper sample conditions [16].

The goal of this study is to investigate the relationship between the 22C3 and SP263 assays used in routine clinical practice and artificially conditioned archival FFPE sections. This research goes on to stress the need of adhering to preanalytical conditions.

## 2. Materials and Methods

### 2.1. Patients and Routine Samples

This study cohort included PD-L1 data from routine consecutive samples of 431 NSCLC between January 2017 and December 2018 and the archives from 314 surgically resected NSCLCs, which were collected from January 2006 to December 2018 from Yeouido St. Mary’s Hospital (Appendix A). The inclusion criteria comprised a diagnosis of NSCLC and nondecalcified specimens [17]. Before the collection of the samples, none of the patients received systemic therapy. Patient data were obtained from electronic medical records. Tumors were classified using the 2015 World Health Organization classification, and staging was done using the AJCC TNM staging (8th edition). This study has been conducted according to principles expressed in the Declaration of Helsinki and is approved by the institutional review board of Yeouido St. Mary’s Hospital (SC18RNSI0075).

### 2.2. Immunohistochemistry Preanalytic Conditions

All biopsy and surgical specimens were preserved in 10% buffered formalin. To prevent cold ischemia, all biopsy specimens were immediately placed in formalin, and all surgical samples were immediately sent to the pathology department and injected with formalin. The routine clinical samples (*n* = 431) were stained simultaneously with PD-L1 22C3 and SP263 assays as a reflex test except for a minority being performed as a serial test. Among the 314 surgical resection archives, there were forty-five 1 year-stored cases, fifty-six 1–3 year-stored cases, one hundred and four 3–6-year stored cases, and one hundred and nine > 6-year stored cases. Fresh paraffin sections from tissue archives were stained with PD-L1 22C3 and SP263 or stained after being preserved at room temperature or 4 °C for 1 week, 2 weeks, and 1 month (Figure 1).

### 2.3. PD-L1 Immunohistochemistry

Formalin-fixed paraffin-embedded (FFPE) fresh tissue blocks were microdissected at 4 µm thick. As a reflex test in the routine 431 cohort, PD-L1 assays were performed on fresh sections as soon as NSCLC is diagnosed. On the other hand, PD-L1 tests were performed on variously prepared paraffin sections from 314 archives. Assays used for staining are PD-L1 clone 22C3 pharmDx kit (Dako, Santa Clara, CA, USA) by Dako Automated Link 48 platform, and SP263 assay (Ventana Medical Systems Inc., Tucson, AZ, USA) by Ventana BenchMark Ultra. The staining process has been done as per the manufacturers’ instructions.

Each case had 5 slides read by two thoracic pathologists: a hematoxylin and eosin (HE) stained slide, one of each 22C3 and SP263 stained slides, and respective negative reagent control slides. Tonsil tissue for 22C3 and placenta for SP263 were used as positive external control in each round of the staining process. The HE slides were checked first to confirm the presence of at least 100 viable tumor cells for adequacy of interpretation. The degree of 22C3 and SP263 PD-L1 expression was examined in compliance with the corresponding scoring system devised for each PD-L1 clone. Partial or complete membranous staining of any intensity, from barely perceptible to strong complete staining, were equally considered positive staining. Only membranous staining of viable tumor cells was counted—cytoplasmic staining of tumor cells or staining of immune cells (including alveolar macrophages), stromal cells, necrotic cells, and cellular debris were disregarded in counting. Negative run controls for each IHC were examined beforehand to consider factors of background staining, such as anthracotic pigmentation, hemorrhage, necrosis, and other artifacts. Two experienced pathologists scored all cases: pathologist A (TJK) and pathologist B (SYK). Each assay was analyzed by both pathologists and the interpretations have been discussed freely to come to an agreement on TPS. The type of assay was not blinded to the pathologists. Both 22C3 and SP263 assays were assessed semi-quantitatively and were reported in tumor proportion score (TPS), calculated by dividing the portion of stained tumor cells by the total number of viable tumor cells. Clinically relevant thresholds of 1% and 50% were established, and the results were divided into three categories: 1% (negative), 1% to 49% (low expression), and ≥50% (high expression).

### 2.4. Statistics

The Chi-square and Fisher’s exact tests were used to assess the relationship between variables and PD-L1 expression. These results were then analyzed by the goodness of fit linear regression for the paired TPS results with Excel (Microsoft, Redmond, WA, USA) and GraphPad Prism 9 (GraphPad Software, La Jolla, CA, USA). Results with *p* < 0.05 were considered statistically significant.

## 3. Results

### 3.1. Characteristics of Routine Clinical Cohort

A total of 431 NSCLC patient cases were reviewed. Table 1 summarizes the demographics of 431 the NSCLC patients. The mean age at diagnosis is 68.6 years. Among 431 patients, 291 (67.5%) are male and 140 (32.5%) female. Ever-smokers counted 216 (50.1%) with a mean pack-years of 19.8 years. Adenocarcinoma comprised most of the cohort (282, 65.4%), followed by squamous cell carcinoma (133, 30.9%), adenosquamous carcinoma (7, 1.6%), and lastly other NSCLC (9, 2.1%): diagnosis consisting of large cell neuroendocrine carcinoma and sarcomatoid carcinoma. Adenocarcinoma histologic type is the only significant factor to show ≥1% TPS for SP263 (p = 0.021). Molecular studies revealed that 101 (23.4%) and 13 (3.0%) cases harbored EGFR and ALK mutations respectively. Two-thirds of the patients are stage III or IV at presentation. The numbers of samples from primary NSCLC acquired by different procedures are as follows: 83 bronchoscopy biopsies (19.3%), 256 percutaneous needle aspiration biopsy (PCNB) (59.4%), and 15 endobronchial ultrasound biopsy (EBUS) (3.5%). Twenty-six biopsies are from metastatic NSCLC (6.0%). Six cytologic specimens with enough aspiration to make cell blocks (1.4%) are included. Surgical resections performed during the data collection period are included as well: 41 lobectomies (9.5%) and 4 wedge resections (0.9%).

### 3.2. Overall Comparison of PD-L1 Assays

The overall concordance rate between 22C3 and SP263 is shown in Table 2. When 22C3 is positive (i.e., TPS ≥ 1%), SP263 is also positive in 92.3%, and 89.9% vice versa. When SP263 shows high PD-L1 expression (TPS ≥ 50%), only 79.6% of 22C3 samples are correspondingly high. On the other hand, when 22C3 shows high PD-L1 expression, SP263 also shows high PD-L1 expression in 94.5%. The adjusted R square (R${}^{2}$) value is calculated as 0.8293.

### 3.3. Comparison of PD-L1 Assays According to Specimen Types

Table 3 shows the discrepancies between 22C3 and SP263 assays when subdivided by specimen type. A high concordance rate is noted in surgical resections including lobectomy (90.2%) and wedge resection (100.0%). On the other hand, a relatively low agreement is shown in biopsy specimens: in increasing order, cell block (66.7%), metastatic site biopsy (69.2%), EBUS (80.0%), PCNB (81.3%), and bronchoscopy biopsy (81.9%). The surgical rection sample’s concordance rate was statistically higher than biopsy (*p* < 0.001). FFPE block age affects concordance among PD-L1 assays. A statistically higher concordance rate was observed in FFPE block age <6 months (*p* = 0.003). The goodness-of-fit is illustrated by a linear regression line between the PD-L1 22C3 and PD-L1 SP263 using interpolation graphs in Figure 2. The R${}^{2}$ value is 83.0% in the whole 431 cohort. The R${}^{2}$ values across histologic types are 80.8%, 86.9%, and 95.3% in adenocarcinoma, squamous cell carcinoma, and the other NSCLC respectively (Figure 2A). Whilst samples from lungs posed an R${}^{2}$ value of 85.0%, those from metastatic sites are less concordant with an R${}^{2}$ value of 68.0% (Figure 2B). The presence of EGFR mutation or ALK rearrangement did not significantly affect the distribution of PD-L1 expression. R${}^{2}$ value of *EGFR* or *ALK* mutated group and both *EGFR* and *ALK wild type* were 0.80 and 0.84, respectively (Figure 2C). Among diagnostic materials, the highest R${}^{2}$ value was noted in surgical specimens with R${}^{2}$ = 95.4%, and the lowest R${}^{2}$ value in cell block R${}^{2}$ = 60.4% (Figure 2D).

### 3.4. Associations of Preanalytic Variables and PD-L1 Staining

To investigate the potential impact of specimen conditions on PD-L1 expression across different assays, we investigated the relationships between FFPE block ages (less than 1 year, 1–3 years, 3–6 years, more than 6 years), FFPE section storage conditions (1 week, 2 weeks, and 1 month) at room temperature or 4 °C (Table 4). Older FFPE blocks were associated with lower PD-L1 expression. TPS greater than 50% of 22C3 (p = 0.007) and SP263 (p = 0.009) compared to TPS less than 1% was considerably more common in FFPE blocks less than 3 years old (*p* = 0.023 and *p* = 0.008, respectively). TPS 1–49 percent of both 22C3 and SP263 was also substantially more common in FFPE blocks aged less than 3 years (p = 0.022 and p = 0.009, respectively). FFPE section storage conditions affected PD-L1 expression. The multivariate model revealed that there was a significantly higher probability of preserved 22C3 expression in recent paraffin section storage in room temperature for TPS 1–49 percent and ≥50 percent (OR = 1.73, *p* = 0.006 and OR = 1.98, *p* = 0.002, respectively), but SP263 showed preserved expression in longer section storage duration in room temperature. A PD-L1 TPS less than 1% as a reference category, PD-L1 TPS 1–49% and TPS greater than 50% were characterized by more recent FFPE block age (less than 3 years) for both 22C3 and SP263 assays and more recent FFPE section (within 1 week) at room temperature for 22C3 assay (Figure 3). A linear regression line between the PD-L1 22C3 and PD-L1 SP263 demonstrates that R square values among four groups of FFPE block age (one year, 1–3 years, 3–6 years, and >6 years) are not significantly different. While the R square values of FFPE sections stored at 4 °C (fresh, 1 week, 2 weeks, and 1 month) are not statistically different, the R square values of FFPE sections stored at ambient temperature declined as storage time increased (Figure 4).

## 4. Discussion

Biomarkers that target the PD-1/PD-L1 axis have become increasingly important in the treatment of patients with advanced NSCLC. PD-L1 expression in pathologic specimens is the only approved biomarker as a treatment determinant at the moment. The variability in assay scoring methods, IHC staining implementations, and thresholds has confounded companion or complementary diagnoses in applying the medication. Although studies cross-examining platforms have shown favorable results [18,19], the current separate diagnostic assays mandate pathologic laboratories to furnish multiple platforms, or else patients will have a limited scope of immunotherapy selection. A specified scoring system requires pathologists to be knowledgeable of different standards and cut-offs. The execution of four different PD-L1 studies for all NSCLC specimens equals a fourfold increase in time and cost. Today’s “one drug – one diagnostic test” calls for the harmonization of disparate assays. Interchangeability of diagnostic materials will be beneficial to the patients, providing grant broader accessibility to immunotherapy, winning time, and saving resources (sparing finite small tumor tissues).

In previous studies, there was an overall consensus that SP142 consistently stains fewer tumor cells in comparison to the four PD-L1 assays. Among the remaining 22C3, SP263, SP142, and 28-8, various correlations have been made. The Blueprint phase 1 [7], the feasibility study, set off this motion by analyzing 38 cases. The Blueprint phase 2 study analyzed 81 lung cancer samples [11]. Another study has shown high hopes for the harmonization between assays by comparing 500 NSCLC samples from commercial sources [12].

One of our study’s strong points is that samples encompass various diagnostic materials. Cell blocks are included as it has been suggested that cytologic material processed to cell blocks demonstrates high agreement in PD-L1 expression when matched with histologic materials [20]. Because cytologic techniques are greatly utilized to gain tissue in lymph nodes and distant metastasis [21], the interchangeability of PD-L1 tests in cell blocks has emerged as a critical therapeutic concern. Despite the sample size restriction, our results demonstrated the lowest concordance between 22C3 and SP263 TPS. The explanations for the variances in the level of expression may be numerous. It has been suggested that the biopsy specimens may not fully represent the correct level of PD-L1 expression [22], explained by intratumoral heterogeneity. Likewise, the two PD-L1 immunostains are not necessarily done on consecutive tissue sections, as it was done with other dozen IHC studies, meaning that two assays may have been assessed on different tumor cell populations within the neoplasm.

Except in a few cases, tumors with EGFR or ALK mutations showed a linear connection between 22C3 and SP263, which is likely attributable to high levels of PD-L1 expression in lung cancer samples with driver mutations like EGFR or ALK mutations [23,24,25]. However, PD-L1 expression in samples from patients with EGFR mutations can be dynamic and varied [26]. Differences in the tumor microenvironment could further explain discrepancies in metastatic site biopsies other than intertumoral heterogeneity. Antigen presentation may be affected by different tumor environments provided by the brain or bone tissue. This necessitates a comparison of PD-L1 staining in other malignancies than NSCLC, which is also in progress of evaluation; for example, in malignant melanoma [27]. Following studies should also address whether there is a discrepancy between 22C3 and SP263 IHC expression in post-treatment samples as the tumor microenvironment shifts. Other limitations include preanalytical conditions before IHC: cold ischemia time, time of fixation, and thickness of sections. In addition, relationships between the sample site and PD-L1 expression have been reported in the literature. The expression of PD-L1 in the primary tumor and lung metastasis, for example, has been found to differ [28,29], and reports of higher PD-L1 expression in the metastasis than in the primary NSCLC [30].

Subjectivity should be taken into account. As previously mentioned, the assessment may be biased due to the earlier introduction of 22C3 IHC, by thus first-hand training with 22C3 stain leads to sensitively in discerning even the faintest membranous staining in 22C3. On the other hand, a paper from Italy analyzed 198 cases of NSCLC archival tissue microarrays and witnessed consistently stronger staining quality in SP263 over 22C3 at both 1% and 50% cutoffs, when antibody staining is run by the appropriate platforms [31].

Strong quality assurance programs for the PD-L1 test are necessary, in addition to attempts to limit possible subjectivity in PD-L1 interpretation. Despite the fact that the majority of analytic procedures have little opportunity for adjustment in assay form, the importance of quality control for preanalytic variables is stronger than ever. The IASLC PD-L1 staining atlas recommends specimens with FFPE blocks older than three years to be discarded [32]. Tissue fixation with 4 percent neutral buffered formalin has been the conventional technique for tissue preservation since the late 19th century when formaldehyde was first used in pathology [16]. Formaldehyde, on the other hand, causes crosslinking and chemical alteration of nucleic acids and proteins, in addition to tissue morphology retention. The complexity and the impact of the chemical cascade is under investigation [33], particularly on PD-L1 expression. The degradation of nucleic acid is observed at FFPE storage for more than 6 months at room temperature [16]. We performed PD-L1 assays on variably prepared FFPE sections from stored archives to determine whether the above-recommended preanalytic factors such as fresh versus archived tissues were relevant. We also used multivariate analysis to confirm the associations between PD-L1 assays and FFPE block ages and FFPE section storage conditions. The antigenicity of PD-L1 22C3 in FFPE sections did not last longer when stored at room temperature. However, FFPE storage at 4 °C showed preserved PD-L1 expression. This result aligns with Hendry et al. in which fresh cut sections from 355 cases on 4-year-old tissue microarray showed only a moderate agreement of tumor cell scores in four PD-L1 assays [34]. Focusing on 22C3 and SP263, a pairwise comparison of the two showed significant differences in tumor cell percentage and in proportion showing ≥1% tumor cell staining, with SP263 consistently staining higher than 22C3, leading to a significantly higher proportion of cases classified as positive. Although studies with TMAs have been limited by nature since TMA has a more restricted representation of the whole tumor, these studies must be interpreted with caution. The authors share the notion of more intense and defined membranous staining in SP263 noted by Munari et al. [31], and this may perhaps give clue to why the antigenicity of SP263 lasted far longer than that of the 22C3 clone.

This is the first study in our knowledge to compare 22C3 and SP263 immunostaining with a large number of fresh specimens directly from a routine diagnostic process in conjunction with variously conditioned old archives. Our findings are consistent with many earlier investigations [7,10,11,12,21,35,36], providing great hopes for a thorough understanding of the interchangeability among different PD-L1 assays.

## 5. Conclusions

Advances in personalized medicine may necessitate re-analysis of tissue collected from a patient several months to years ago using cutting-edge molecular techniques. As demonstrated here, storing FFPE blocks at room temperature in normal archives results in discordance of 22C3 and SP263. Because of such discordance, at some point, FFPE blocks may become unsuitable for research or diagnostic retesting. Due to rising costs, cooled storage of all tissue samples accumulated in a normal pathology laboratory may not be practical. However, from a prospective point of view, whenever high-quality molecular information is required, the use of formaldehyde preserving FFPE blocks or at least FFPE sections at 4 °C or freezing should be considered, particularly for 22C3 tests. Otherwise, SP263 may be a preferable option for room temperature storage. Understanding the various ideal settings for each assay will improve PD-L1 testing quality.

## Figures and Tables

**Figure 1 cancers-14-03138-f001:**
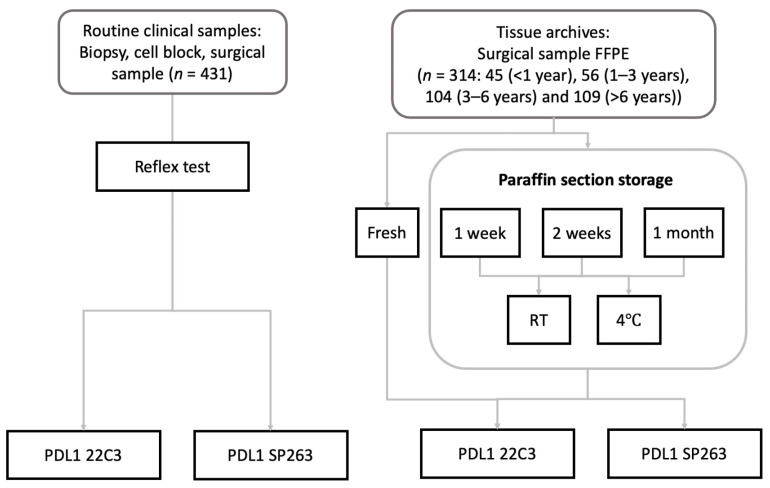
Schematic overview of the study setup. routine clinical samples (*n* = 431) were stained simultaneously with PD-L1 22C3 and SP263 assays as s reflex test. Fresh paraffin sections from tissue archives (*n* = 314): 45 (1 year), 56 (1–3 years), 104 (3–6 years), and 109 (>6 years) were stained with PD-L1 22C3 and SP263 or stained after being preserved at room temperature or 4 °C for 1 week, 2 weeks, and 1 month.

**Figure 2 cancers-14-03138-f002:**
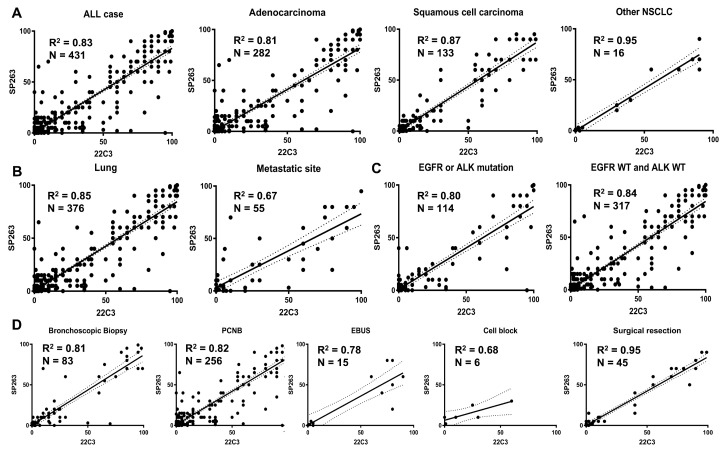
Goodness of fit linear regression between PD-L1 22C3 and SP263 assay. (**A**) Histologic types: all case, Adenocarcinoma, Squamous cell carcinoma and other NSCLC. (**B**) The tested site: lung and metastatic site. (**C**) Mutation status: *EGFR* or *ALK* mutation and both *EGFR* and *ALK wild type*. (**D**) Sample type: bronchoscopic biopsy, percutaneous needle biopsy, EBUS, cell block and surgical resection.

**Figure 3 cancers-14-03138-f003:**
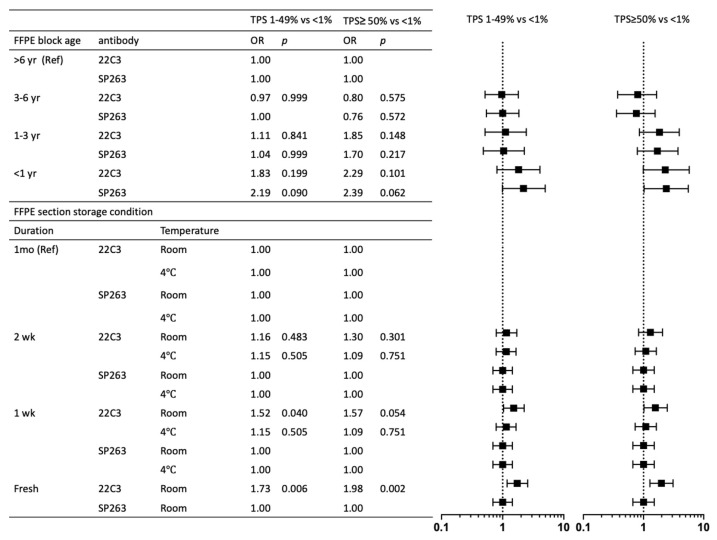
Forest plot of the OR (with 95% confidence intervals) of having PD-L1 tumor proportion score of 1% to 49% and greater than 50% compared to less than 1% according to the FFPE block age and the FFPE section storage conditions (multivariate analyses). OR: odds ratio; Ref: reference; yr: year; wk: week.

**Figure 4 cancers-14-03138-f004:**
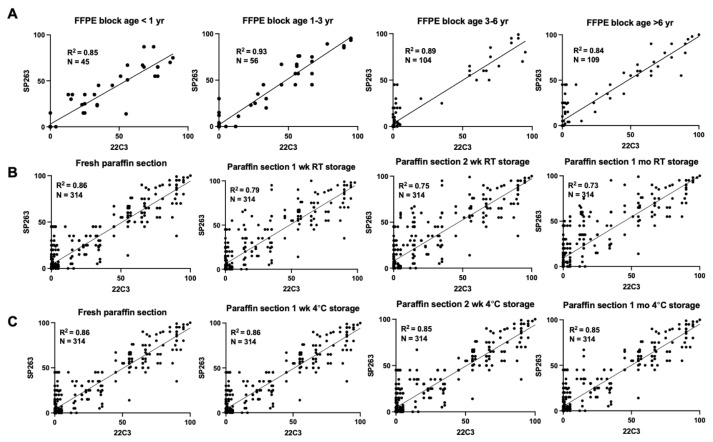
Goodness of fit linear regression between PD-L1 22C3 and SP263 assay in 314 conditioned archives. (**A**) Paraffin block divided in four categories: <1 year, 1–3 year, 3–6 year, and >6 year. (**B**) Paraffin section stored as fresh, 1 week, 2 week and 1 month at room temperature. (**C**) Paraffin section stored as fresh, 1 week, 2 week and 1 month at 4 °C. FFPE: formalin fixed paraffin embedded; RT: room temperature; yr: year; wk: week.

**Table 1 cancers-14-03138-t001:** Demographics of routinely performed PD-L1 tests in 431 patients with NSCLC.

		22C3 TPS ≥ 50%		SP263 TPS ≥ 50%		22C3 TPS ≥ 1%		SP263 TPS ≥ 1%	
		(*n* = 108)		(*n* = 91)		(*n* = 307)		(*n* = 299)	
**Varable**	**Total (%)**	* **n** * **(%)**	* **p** *	* **n** * **(%)**	* **p** *	* **n** * **(%)**	* **p** *	* **n** * **(%)**	* **p** *
Age, mean (SD)	68.6 (10.3)	67.9 (8.7)	0.258	67.8 (9.0)	0.248	68.4 (9.9)	0.377	68.2 (10.3)	0.294
Sex									
Male	291 (67.5)	74 (68.5)	0.179	61 (67.0)	0.919	214 (69.7)	0.158	208 (69.6)	0.182
Female	140 (32.5)	34 (31.5)		30 (33.0)		93 (30.3)		91 (30.4)	
Smoking									
Never smoker	215 (49.9)	51 (47.2)	0.598	46 (50.5)	0.89	154 (50.2)	0.94	149 (49.8)	1
Ever smoker	216 (50.1)	57 (52.8)		45 (49.5)		153 (49.8)		150 (50.2)	
Histology									
ADC	282 (65.4)	69 (63.9)	0.786	56 (61.5)	0.380	193 (62.9)	0.099	185 (61.9)	**0.021**
SCC	133 (30.9)	33 (30.6)		29 (31.9)		102 (33.2)		102 (34.1)	
other NSCLC	16 (3.7)	6 (5.6)		6 (6.6)		12 (3.9)		12 (4.0)	
Mutation									
*EGFR*	101 (23.4)	18 (16.7)	0.306	14 (15.4)	0.276	68 (22.1)	0.371	68 (22.7)	0.813
*ALK*	13 (3.0)	6 (5.5)		6 (6.6)		9 (2.9)		10 (3.3)	
*EGFR ALK WT*	317 (73.5)	84 (77.8)		71 (78.0)		230 (74.9)		221 (73.9)	
Stage									
I–II	148 (34.3)	33 (30.7)	0.970	30 (32.9)	0.818	97 (31.6)	0.576	97 (32.4)	0.825
III	176 (40.9)	50 (46.2)		41(45.1)		134 (43.6)		140 (43.5)	
IV	107 (24.8)	25 (23.1)		20 (22.0)		76 (24.8)		72 (24.1)	
Sample									
Biopsy	386 (89.6)	95 (88)	0.656	78 (85.7)	0.247	279 (91)	0.503	269 (90)	0.733
Surgical resection	45 (10.4)	13 (12)		13 (14.3)		28 (9)		30 (10)	

Abbreviations: EGFR: epidermal growth factor receptor; ALK: anaplastic lymphoma kinase; ADC: adenocarcinoma; SCC: squamous cell carcinoma; NSCLC: non-small-cell lung cancer; WT: wild type. Values in bold are statistically significant; Age was evaluated by Chi square test and variables other than age are evaluated by *t*-test; Histology was evaluted as adenocarcinoma versus all others; Mutation was evaluated as EGFR or ALK mutation versus both EGFR ALK wild type; Stage was evaluated as stage I–II versus III–IV; ^*f*^ biopsy versus surgical resection.

**Table 2 cancers-14-03138-t002:** Overall concordance rate between 22C3 and SP263 in routine clinical samples (N = 431).

Assay	22C3	TPS ≥ 50%	SP263	TPS ≥ 50%	22C3	TPS ≥ 1%	SP263	TPS ≥ 1%
22C3	108/108	100%	86/91	94.50%	307/307	100%	276/299	92.30%
SP263	86/108	79.60%	91/91	100%	276/307	89.90%	299/299	100%

Abbreviations: TPS: tumor proportion score.

**Table 3 cancers-14-03138-t003:** Distribution of PD-L1 22C3 and SP263 according to specimen type in routine clinical samples.

		Concordant	Discordant						
		22C3 TPS	1–49%	<1%	≥50%	<1%	≥50%	1–49%	
Sample N (%)	Total	SP263 TPS	<1%	1–49%	<1%	≥50%	1–49%	≥50%	*p*
Bronchoscopic biopsy	83	68 (81.9)	7 (8.4)	3 (3.6)	0	0	3 (3.6)	2 (2.4)	**<0.001**
PCNB	256	208 (81.3)	20 (7.8)	14 (5.5)	0	0	12 (4.7)	2 (0.8)	
Cell block	6	4 (66.7)		1 (16.7)	0	0	1 (16.7)		
EBUS	15	12 (80.0)	1 (6.7)		0	0	2 (13.3)		
Metastatic site biopsy	26	18 (69.2)	2 (7.7)	2 (7.7)	0	0	3 (11.5)	1 (3.8)	
Lobectomy	41	37 (90.2)	1 (2.4)	3 (7.3)	0	0			
Wedge resection	4	4 (100.0)			0	0			
FFPE block age									
<6 mo	418	349	31	16	0	0	22	5	**0.003**
≥6 mo	13	6	0	7	0	0	0	0	
Total	431	351	31	23	0	0	21	5	

Abbreviations: TPS: tumor proportion score; PCNB: percutaneous needle biopsy; EBUS: endobronchial ultrasound biopsy; FFPE: formalin-fixed, paraffin-embedded tissue. Values in bold are statistically significant; *p*-values are evaluated by Fisher’s exact test; concordance of PD-L1 assays between sample types (biopsy versus surgical resection) and concordance of PD-L1 assays between FFPE block ages (<6 mo versus ≥6 mo).

**Table 4 cancers-14-03138-t004:** PD-L1 tumor proportion score in the 314 conditioned archives.

			PD-L1 TPS			
FFPE Block Age	Assay		<1%	1–49%	≥50%	*p*
<1yr	22C3		17	15	13	0.224 ${}^{a}$
1-3 yr	22C3		26	14	16	**0.007** ${}^{b}$
3-6 yr	22C3		60	28	16	
>6 yr	22C3		60	29	20	
<1yr	SP263		15	17	13	0.184 ${}^{c}$
1-3 yr	SP263		26	14	16	**0.009** ${}^{d}$
3-6 yr	SP263		58	30	16	
>6 yr	SP263		58	30	21	
FFPE section storage condition						
Duration		Temp				
Fresh	22C3	RT	163	86	65	**0.004** ^ *e* ^
1 wk	22C3	RT	177	81	56	**0.033** ${}^{f}$
2 wk	22C3	RT	195	68	51	0.469 ${}^{g}$
1 mo	22C3	RT	209	63	42	**0.002** ${}^{h}$
1 wk	22C3	4 °C	169	84	61	**0.046** ${}^{i}$
2 wk	22C3	4 °C	169	84	61	0.253 ${}^{j}$
1 mo	22C3	4 °C	178	77	59	0.467 ${}^{k}$
Fresh	SP263	RT	157	91	66	0.467 ${}^{l}$
1 wk	SP263	RT	157	91	66	0.8 ${}^{m}$
2 wk	SP263	RT	157	91	66	0.8 ${}^{n}$
1 mo	SP263	RT	157	91	66	1 ${}^{o}$
1 wk	SP263	4 °C	157	91	66	
2 wk	SP263	4 °C	157	91	66	
1 mo	SP263	4 °C	157	91	66	

Abbreviation: TPS: tumor proportion score; yr: year; wk: week; mo: month; Temp: temperature; RT: room temperature. Values in bold are statistically significant. ^*a*^ PD-L1 22C3 1–49% versus <1% according to FFPE block age (<3 yr versus ≥3 yr). ^*b*^ PD-L1 22C3 ≥50% versus <1% according to FFPE block age (<3 yr versus ≥3 yr). ^*c*^ PD-L1 SP263 1–49% versus <1% according to FFPE block age (<3 yr versus ≥3 yr). ^*d*^ PD-L1 SP263 ≥50% versus
<1% according to FFPE block age (<3 yr versus ≥3 yr). ^*e*^ PD-L1 22C3 1–49% versus <1% according to FFPE section
storage duration (fresh versus 1 mon) in RT. ^*f*^ PD-L1 22C3 1–49% versus <1% according to FFPE section storage
duration (1 wk versus 1 mon) in RT. ^*g*^ PD-L1 22C3 1–49% versus <1% according to FFPE section storage duration
(2 wk versus 1 mon) in RT. ^*h*^ PD-L1 22C3 ≥50% versus <1% according to FFPE section storage duration fresh
versus 1 mon in RT. ^*i*^ PD-L1 22C3 ≥50% versus <1% according to FFPE section storage duration (1 wk versus
1 mon) in RT. ^*j*^ PD-L1 22C3 ≥50% versus <1% according to FFPE section storage duration (2 wk versus 1 mon)
in RT. ^*k*^ PD-L1 22C3 1–49% versus <1% according to FFPE section storage duration (1 wk versus 1 mon) at 4 °C.
^*l*^ PD-L1 22C3 1–49% versus <1% according to FFPE section storage duration (2 wk versus 1 mon) at 4 °C. ^*m*^ PD-L1
22C3 ≥50% versus <1% according to FFPE section storage duration (1 wk versus 1 mon) at 4 °C. ^*n*^ PD-L1 22C3
≥50% versus <1% according to FFPE section storage duration (2 wk versus 1 mon) at 4 °C. ^*o*^ PD-L1 SP263 showed
no expression difference in various FFPE section storage conditions.

## Data Availability

Not applicable.

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
