# Peer review of "Comprehensive Comparison of 22C3 and SP263 PD-L1 Expression in Non-Small-Cell Lung Cancer Using Routine Clinical and Conditioned Archives"

_cancers, 2022, doi:10.3390/cancers14133138_

Round 1

Reviewer 1 Report

1. The manuscript needs some careful edits. For instance, "22C3 showed highest concordance rate with 94.5% of SP263  tumor proportion score (TPS) ≥50% and 98.9% 78.8% of SP263 TPS 1-49%, while SP263 showed  concordance rate with 79.6% of 22C3 TPS ≥50% and 82.4% of 22C3 TPS 1-49%." "FFPE blocks older than 3 years and sections stored at room temperature for more than 1 week may warrant underestimate of PD-L1 status".

2. Figure 2 shows the correlation regarding cancer types and other factors. It is interesting to see the changes regarding tumor mutations. Is there a correlation between PD-L1 levels and these mutations? Any potential reasons for the difference detected by the two assays?

3. For the data shown in Tables 4 and 5, how about the linear correlation? Huge difference?

Author Response

Dear reviewer,

Thank you for the precious comments

1. The manuscript needs some careful edits. For instance, "22C3 showed highest concordance rate with 94.5% of SP263  tumor proportion score (TPS) ≥50% and 98.9% 78.8% of SP263 TPS 1-49%, while SP263 showed  concordance rate with 79.6% of 22C3 TPS ≥50% and 82.4% of 22C3 TPS 1-49%." "FFPE blocks older than 3 years and sections stored at room temperature for more than 1 week may warrant underestimate of PD-L1 status".

-> We checked typos and errors and corrected them all.

2. Figure 2 shows the correlation regarding cancer types and other factors. It is interesting to see the changes regarding tumor mutations. Is there a correlation between PD-L1 levels and these mutations? Any potential reasons for the difference detected by the two assays?

There are several reports tumot mutations are associated with high level of PD-L1 expression, I added those contents in discussion. 

3. . For the data shown in Tables 4 and 5, how about the linear correlation? Huge difference?

Thank you for the good point. we added figure4 for showing correlation in archival samples

Reviewer 2 Report

The manuscript describes a well-designed study on two different commercial assays to evaluate PDL1 expression in tumor samples. The study compares both assays over a large collection of lung cancer tumor biopsies, processed and stored under different conditions. The study, although highly technical, is of interest for both basic and clinical researchers. It is well-written, with minor spelling corrections required. It reads well and it is well presented. It is my opinion that the manuscript is suitable for publication.

Author Response

Thank you for the precious comments

We corrected typos and errors and added more figure for the clarification of the manuscript.